# Effect of Magnetized Freezing Extender on Membrane Damages, Motility, and Fertility of Boar Sperm Following Cryopreservation

**DOI:** 10.3390/ani13040634

**Published:** 2023-02-11

**Authors:** Seunghyung Lee, Yong-Min Kim, Hee-Tae Cheong, Choon-Keun Park, Sang-Hee Lee

**Affiliations:** 1College of Animal Life Sciences, Kangwon National University, Chuncheon 24341, Republic of Korea; 2Swine Science Division, National Institute of Animal Science, Rural Development Administration, Cheonan 31000, Republic of Korea; 3College of Veterinary Medicine, Kangwon National University, Chuncheon 24341, Republic of Korea

**Keywords:** boar sperm, cryopreservation, freezing extender, magnetization

## Abstract

**Simple Summary:**

Understanding freezing is essential for stable cryopreservation of sperm. In the freezing process, ice crystal is developed from a change of matter from liquid water to solid water. Unfortunately, they tend to make damage of membrane in frozen sperm. From a perspective, sperm membrane damage due to ice crystal formation, magnetized freezing water that has easy supercooling, and the construction of smaller ice crystals because weaken the hydrogen bonds between the water molecules also has an infinite potential to cryopreserve sperm. Here, we applied the magnetization technique for making of freezing extender, and the extenders were used in the freezing process in sperm cryopreservation. As a result, sperm membrane damage was reduced in frozen sperm by a magnetized freezing extender, and we determined that in vitro fertility of frozen sperm by magnetized extender was improved. In conclusion, the magnetization technique would be useful for sperm cryopreservation. Furthermore, the study suggests that the technique has great potential for storing frozen material for biological aspects.

**Abstract:**

Magnetized water is defined as the amount of water that has passed through a magnet. The magnetic field weakens the hydrogen bonds between the water molecules, leading to the magnetized liquid acquiring special characteristics such as easy supercooling and forming smaller ice crystals. We researched the influences of a magnetized freezing extender on cell membrane damage and in vitro fertilization of boar sperm during cryopreservation. The freezing extenders were passed through 0, 2000, 4000, and 6000 gausses (G) of magnetic devices using a liquid cycling pump system and then used for the sperm freezing process. The damage to plasma, acrosomal, and mitochondrial membranes in frozen-thawed spermatozoa was investigated by flow cytometry, and motility was assessed using the CASA system. The fertility of frozen-thawed sperm was estimated using in vitro fertilization. The damage to the membranes was significantly decreased in the magnetized freezing extender by the 6000 G magnetic field compared to that of the control in frozen-thawed sperm, and motility was increased in the 6000 G group. Although there were no significant differences in the cleavage rates of in vitro fertilized oocytes among the treatment groups, the ratio of blastocyst formation increased in the magnetized freezing extender groups compared with that in the control group. The number of blastocysts was significantly higher in the 4000 G group than in the 0 G group. In conclusion, these results suggest that a magnetized freezing extender could improve the freezability of sperm and the development of oocytes fertilized in vitro with frozen-thawed sperm.

## 1. Introduction

Sperm cryopreservation has various advantages, such as long-term storage, and is a powerful tool for preserving and restoring endangered species. Frozen semen is widely used to improve the preservation of superior breeds and the fertility of domestic animals. It can extend the availability of sperm for artificial insemination (AI), in vitro fertilization (IVF), and biological experiments, irrespective of time or location [1,2]. However, the formation of ice crystals [3] during freezing affects the plasma, acrosomal, and mitochondrial membranes of frozen-thawed sperm [4,5,6]. Eventually, this cryo-damage leads to a reduction in viability, motility, and fertility of the sperm [7,8,9].

Cryo-damage by ice crystals increases the plasma membrane’s permeability and disruption, leading to sperm death [10]. Ice crystal formation is one of the main sources of physical damage to sperm membranes [11]. Some studies have reported that ice crystal formation and uniform nucleation affect freezability in boar sperm [12,13]. Thus, a more uniform ice crystal during freezing can minimize cryo-damage in the plasma membrane, inner and outer acrosomal membranes, and mitochondrial membrane in sperm.

Because water is the main material used in the freezing extender, understanding the congelation of water is essential for cryopreservation for biological purposes [14]. In the freezing process, ice crystals are developed from the unbound water within the lattice space of the material and the water within each cell (intracellular freezing). For long-time preservation, the biological product is to be frozen [15]. These ice crystals tend to grow into relatively large crystals within the material, thus, damaging the structure by rupturing the cell walls and hence the integrity of the frozen material [15]. Eventually, the formation of ice crystals in the lattice space negatively influences the cellular components; thus, it is important to control the size of the ice crystals during freezing.

Magnetized water is defined as water that passes through a magnet (general magnet or electromagnet). It can ionize and activate water molecules changing their structure hexagonal, like the water in our body. Magnetic fields weaken the van der Waals bonding between water molecules [16] and compared to general water, magnetized water has the special characteristic of easy supercoiling [17], producing smaller ice crystals, a high electronic donor, an increase in electric conduction [18], and an increase in hexamer structure [18]. In practice, some studies reported that a magnetized freezing extender improved frozen-thawed sperm in humans [19], rabbits [20], chickens [21], and sheep [22] but a study on magnetized freezing extender has not yet been reported on the cryopreservation of boar sperm. Therefore, this study aimed to investigate the effect of the magnetized freezing extender on sperm characteristics and the development of oocytes in vitro fertilized with cryopreserved boar semen.

## 2. Materials and Methods

### 2.1. Sample Preparation

A total number of 25 ejaculates (5 ejaculates per boar) were collected from five pigs using the gloved-hand method once a week (Gumbo Artificial Insemination Industry, Wonju, Republic of Korea). The age of the experimental five pigs (Duroc) is 25.6 ± 4.2 months. The semen was diluted with modified Modena B, which was manufactured according to a previous study [23]. Only ejaculates with more than 80% total motility, 70% plasma membrane integrity, and 20% acrosome membrane damage were used for the experiment, and the samples were transported to the laboratory within 2 h at 18 °C [23,24].

### 2.2. Magnetization of Freezing Extender

Freezing extender and cryopreservation were performed as previously described [25]. The 1st Lactose Egg-Yolk freezing extender (LEY-1) comprised 11% α-lactose (Sigma, St. Louis, MO, USA) and 20% egg yolk. Then the 2nd freezing extender (LEY-2) was made from LEY-1 supplemented 9.0% glycerol (Sigma) and 1.5% Orvus Es Paste (Nova Chemical Sales Inc., Calgary, Alberta, Canada). The magnet devices for the magnetization of LEYs were manufactured by 2000 gauss (G) (Figure 1A, yellow arrow) and 4000 G (Figure 1A, red arrow) neodymium magnets, and plastic pipes were placed to flow LEYs into magnetic devices (Figure 1B). The 2000 G and 4000 G fields for the flow of LEYs were created by regulating the space between two 2000 G (Figure 1C, yellow arrow) and 4000 G (Figure 1D, red arrow) neodymium magnets. The 6000 G field was made using a designed stainless device by auto computer-aided design (AutoCAD), and the space for LEYs was 7.0 mm. Two 6000 G neodymium magnets (Figure 1E, red arrows) were attached outside the stainless device [26]. The spaces for the flow of LEYs into the magnetic field were measured using a gauss meter (Tesla Meter TM-701, KANETEC, Tokyo, Japan). The LEYs were circulated passing through 0, 2000, 4000, and 6000 G magnetic fields using a peristaltic pump (BT100-2J, Longerpump, Baoding, China) at 18 °C (LEY-1) and 5 °C (LEY-2) for 10 min at 100 rpm before cryopreservation (Figure 1F).

### 2.3. Sperm Freezing and Thawing

The semen sample was centrifuged to remove the supernatant and seminal plasma (410× *g*, 5 min). Then the fractions of sperm fraction were used for freezing. And experimental tubes containing sperm diluted with magnetized LEY-1 (18 °C) were put into 500 mL of 18 °C water, and then, they were cooled until 5 °C in a −18 °C freezer for 120 min. Next, the sample was diluted with magnetized LEY-2 (5 °C, 50% of 1st freezing extender volume). The final concentration of sperm was 1.0 × 10^9^ sperm/mL. The sample was packed in 0.25 mL-straw (a refrigerated room, 5 °C), and pre-frozen (−120 °C, 10 min) and cryopreservation (−196 °C). The 3 straws per sample were thawed (38 °C, 45 s) and centrifuged (410 g, 5 min). The freezing extender was carefully removed, and resuspended with Modena B and centrifuged (410 g, 5 min). After the supernatant was removed, the sample was resuspended in phosphate-buffered saline (PBS) to 7.5 × 10^6^ sperm/mL for detection of plasma membrane integrity, damaged acrosomal membrane, and mitochondrial membrane integrity and in modified Modena B for analysis of motility.

### 2.4. Plasma Membrane Integrity, Damaged Acrosomal Membrane, and Mitochondrial Membranes Integrity

To detect plasma membrane integrity, the samples were diluted with 40 nM SYBR-14 (Molecular Probes, Eugene, OR, USA), and incubated with 2.0 µM propidium iodide (PI; Sigma) at 38°C for 5 min, then centrifuged (410× *g*, 5 min). After removing the supernatant, the frozen-thawed sperm were resuspended in 1 mL of PBS [25]. Because acrosomal membrane damage induces acrosome reaction, the damaged acrosomal membrane was detected according to a previous study [23]. The samples were treated with 2.0 µM lectin from *Arachis hypogaea* (FITC-PNA; Sigma) and 2.0 µM PI, which were incubated at 38 °C for 5 min, then centrifuged at 410× *g* for 5 min and resuspended in 1.0 mL PBS. Mitochondrial membrane integrity is directly related to mitochondrial damage. We conduct this according to a previous study [23]. To detect mitochondrial membrane integrity, the samples were diluted with 2.4 µM Rhodamine123 (Rho, Sigma) and 2.0 µM PI, then incubated (10 min, 38 °C). The stained sample was centrifuged (410× *g*, 5 min), and after removing the supernatant, 1.0 mL PBS was resuspended. Then, 10,000 sperm from all samples were analyzed by flow cytometry (488 nm, FACSCalibur, BD Biosciences, Franklin Lakes, NJ, USA). Dot plot analysis condition settings regarding forward scatter (FSC), side scatter (SSC), FL-1, and FL-2 were carried out according to a previous study [23], and the flow cytometry data were analyzed using CELLQuest software (Version 6.0, BD). Plasma membrane intact sperm were stained SYBR14+/PI− (Figure 2A, red box), and FITC-PNA positive sperm (FITC-PNA+/PI− and FITC-PNA+/PI+) were sperm of damaged acrosomal membrane (Figure 2B, red box). Mitochondrial membrane integrity was Rho positive and PI negative sperm (Rho+/PI−).

### 2.5. Sperm Motility

Sperm motility was assessed using the computer-assisted sperm analysis (CASA) system (Hamilton Thorne, Beverly, MA, USA) according to a previous study [27]. For approximately 200 sperm samples, sperm motility was measured at 10 × objective phase-contrast mode. 10 μL of sperm diluted to a concentration of 1.0 × 10^6^ sperm/mL was mounted in a slide glass (IMV Technologies, L’ Aigle, France). Five microscopic fields-selected sections were randomly scanned and analyzed for sperm motility. Sperm parameters including total motility (%), progressive motility (%), velocity average path (μm/s), velocity straight line (μm/s), and curvilinear velocity (μm/s) were analyzed using the CASA program.

### 2.6. In Vitro Fertilization

In vitro fertilization (IVF) experiment was assessed as described previously [23]. To investigate fertility, cryopreserved sperm samples using a magnetized freezing extender with different magnetic intensities (0, 2000, 4000, and 6000 G) were used for IVF. 50 to 60 oocytes were used to evaluate in vitro fertility of frozen-thawed sperm. Blastocysts were stained for the measurement of cell number using 1.0 μg/mL Hoechst 33342 for 30 min in a dark room and detected using a fluorescence microscope (Olympus, Tokyo, Japan).

### 2.7. Statistical Analyses

The data were analyzed using SAS ver. 9.4 software (SAS Institute, Cary, NC, USA). Data are presented as the mean ± standard error. Sperm membrane integrity, damages, and in vitro fertility data distribution and homogeneity of variance were checked using Leven’s tests. Data were evaluated by analysis of variance and Duncan’s multiple range test using general linear models. A *p* value < 0.05 was considered to indicate statistical significance.

## 3. Results

### 3.1. Effect of Magnetized Freezing Extender on Plasma Membrane Integrity in Frozen-Thawed Boar Sperm

Figure 2A shows the state of plasma membrane integrity during cryopreservation of semen frozen with a magnetized freezing extender. The proportion of intact plasma membrane was higher (*p* < 0.05) in 4000 G (58.8 ± 1.2%) and 6000 G (61.7 ± 2.0%) than in 0 G (52.3 ± 1.2%) and 2000 G (53.9 ± 1.3%) (Figure 2D).

### 3.2. Effect of Magnetized Freezing Extender on the Damaged Acrosomal Membrane in Frozen-Thawed Boar Sperm

Changes in damaged acrosomal membranes during cryopreservation of semen diluted with a magnetized freezing extender are shown in Figure 2B. The proportions of the damaged acrosomal membrane were significantly (*p* < 0.05) lower in the 4000 G (32.2 ± 1.8%) and 6000 G (30.4 ± 2.5%) magnetized freezing extender groups than in 0 G (40.8 ± 1.0%) and 2000 G (38.7 ± 1.3%) treatment groups (Figure 2E).

### 3.3. Influence of Magnetized Freezing Extender on Mitochondrial Membrane Integrity in Frozen-Thawed Boar Sperm

In Figure 2C, the changes in the integrity of the mitochondrial membrane integrity show during cryopreservation in the diluted semen with a magnetized freezing extender. However, there were no significant differences between the treatment groups (0, 2000, and 4000 G, Figure 2F). All sperm proportion with damaged mitochondria damage was significantly (*p* < 0.05) lower in the 6000 G group (88.5 ± 1.3%) freezing extender than in the 0 G (79.7 ± 3.0%) and 2000 G (82.0 ± 1.4%) treatment groups (Figure 2F).

### 3.4. Changes of Frozen-Thawed Sperm Motility Using Magnetized Freezing Extender in Pigs

Table 1 shows the changes in motility in frozen-thawed sperm using a magnetized freezing extender. Total motility was increased in 6000 G treatment groups (42.1 ± 1.6%) than in other treatments (*p* < 0.05). However, there were no significant differences between the groups in the 0, 2000, and 4000 G treatment groups in PM, VAP, VSL, and VCL.

### 3.5. Formation of Embryo and Blastocyst Rates and Total Cell Numbers of Blastocyst

Table 2 shows the cleavage rate and blastocyst formation of the oocyte at 168 h after IVF following sperm cryopreservation in the magnetized freezing extender are summarized. The oocyte cleavage rate using sperm preserved for various amounts of time did not significantly differ among the differently magnetized groups (0, 2000, 4000, and 6000 G). During cryopreservation, the blastocyst formation rate was significantly higher in the 2000 G and 4000 G freezing extender groups than in the 0 G treatment group (*p* < 0.05). The total cell number of blastocyst was significantly higher (*p* < 0.05) in 4000 G (*n* = 15, 58.2 ± 2.4) and 6000 G (*n* = 17, 59.1 ± 2.1) treatment groups than in 0 G (*n* = 14, 51.1 ± 2.0) and 2000 G (*n* = 16, 54.7 ± 1.7%) treatment groups (Figure 3).

## 4. Discussion

AI contributes to genetic improvements and has been applied extensively in domestic industries worldwide [28]. The development of AI techniques has accelerated andrology, such as sperm physiology, storage methods, cryopreservation, and our understanding of seminal plasma composition for successful fertilization [1,26,29]. In the swine industry, a large amount of semen from a single male pig is used for AI in several pigs, and liquid and freezing techniques are widely used for the long-term preservation of transportation to individual swine farms [30]. For efficient preservation, ejaculated boar semen must be rapidly mixed with an extender for long-term transport and preserved for the long term [31]. Previous studies have reported that a magnetized semen extender protects boar sperm membranes of plasma, acrosomes, and mitochondria [23] and has antioxidative ability during liquid preservation [26]. These results suggest that the increased electronic donor of water [18] of magnetized semen extender [23] decreases free radicals, leading to improved antioxidative ability [26]. Similarly, our results showed that magnetized freezing extenders protected sperm membranes, such as the plasma, acrosome, and mitochondria. This means that the increased electronic donor of the magnetized freezing extender protected against cryo-damage by ice crystal formation during freezing.

Somatic cells recover organelles and produce protective proteins against negative stimulation by the central dogma, but sperm have little cytoplasm and do not have mitosis ability; hence, the sperm is vulnerable to external damage such as chemical and physical damage [32,33]. In particular, physical damage from ice crystal formation during freezing directly affects plasma and outer acrosomal membranes, resulting in excessive acrosome reaction and injury of the mitochondrial membrane, resulting in low energy production for motility [4]. The ice crystal area and space are related to the plasma membrane and damage to the acrosome status in frozen boar sperm [12,13]. Thus, understanding the mechanism of ice crystal shape and size formation for freezing can be the main topic to improve cryopreservation in terms of protecting sperm organelles such as the plasma membrane, acrosome, and mitochondria from physical damage. Eventually, these organelles were protected by a magnetized freezing extender during cryopreservation, and it would be beneficial to motility in frozen-thawed boar sperm.

The final size, shape, and distribution of the resultant ice crystals depend on the cooling rate, degree of supercooling, and nucleation temperature [15,34,35]. In practice, cooling and freezing rates influence viability and motility in frozen-thawed boar sperm [36], in this study, we chose optimal cooling and freezing condition for cryopreservation of boar sperm according to the previous study. The magnetic field influences the vibration between the water molecules, which reduces crystallization and results in smaller, less clustered, and more uniform crystals [15,37]. The magnetic field reduces damage from ice crystals; eventually, it can be a useful tool to improve the preservation of biological materials. In a previous study, electromagnetic waves decreased water cluster size, surface tension, viscosity, and density of freezing media in human sperm, the magnetized freezing media improved viability, plasma membrane integrity, number of intact acrosomes, and mitochondrial membrane potential in frozen-thawed human sperm [19]. In particular, a study on electro-magnetized freezing media reported that small water clusters in electro-magnetized freezing media seem to lead to the formation of smaller ice crystals, which positively affect frozen-thawed sperm characteristics. In practice, a magnetized freezing extender improved sperm characteristics during cryopreservation in rabbits [20], chickens [21], and sheep [22], which was probably due to the reduction of ice nucleation of crystals in freezing media treated with magnetization. Additionally, our study showed that damage to plasma, acrosomal, and mitochondrial membranes was decreased in freezing extenders treated with 4000 and 6000 G neodymium magnets, motility was increased in the 6000 G treatment group, and the formation and cell number of blastocysts were improved in in vitro fertilized oocytes with frozen-thawed boar sperm. Similarly, some studies reported that magnetization techniques increased the litter size and litter weight of rabbit sperm [20] and the fertility and hatching rate of frozen-thawed rooster sperm [21]. Although the magnetized freezing extender did not improve the oocyte cleavage of frozen-thawed boar sperm in our results, we found that frozen-thawed sperm by magnetized freezing extender could improve the embryo quality in in vitro fertilized oocytes. This suggests that the magnetized freezing extender inhibits ice nucleation at higher degrees of supercooling, inducing delays in the initiation of ice nucleation [15], which leads to cryo-abilities such as in vitro fertility and membrane integrity in frozen-thawed sperm.

## 5. Conclusions

These results show that a magnetized freezing extender increases plasma membrane integrity, reduces acrosomal membrane, and improves mitochondrial membrane integrity and motility of frozen-thawed boar sperm. The blastocyst formation rates and total cell number of in vitro fertilized oocytes were enhanced by freeze-thawing with the magnetized freezing extender. Therefore, magnetized freezing extenders treated with 4000 and 6000 G magnetic fields would be useful for the cryopreservation of sperm in pigs.

## Figures and Tables

**Figure 1 animals-13-00634-f001:**
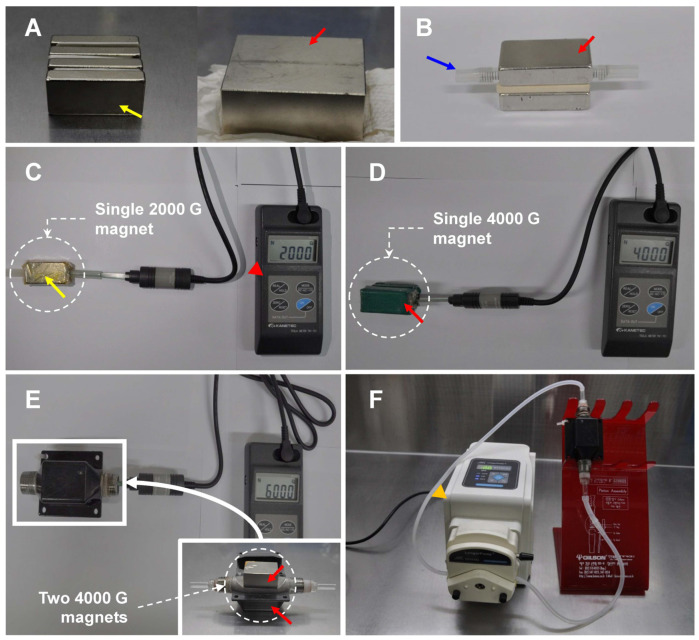
Neodymium magnetic equipment for production magnetized culture medium and freezing extender, neodymium magnet of 2000 G (yellow arrows) and 4000 G (red arrows) magnetic intensity (**A**), installed plastic pipe (blue arrow) for flowing extender (**B**), 2000 G (**C**) and 4000 G (**D**) magnetic intensity were formed by single neodymium magnet equip, and measured (a magnetometer, red arrowhead). Two 4000 G neodymium magnets (red arrows) were used for the formation of 6000 G magnetic intensity (**E**). The culture medium and freezing extender were rotated using a peristaltic pump (yellow arrowhead) for magnetization in neodymium magnetic equipment (**F**).

**Figure 2 animals-13-00634-f002:**
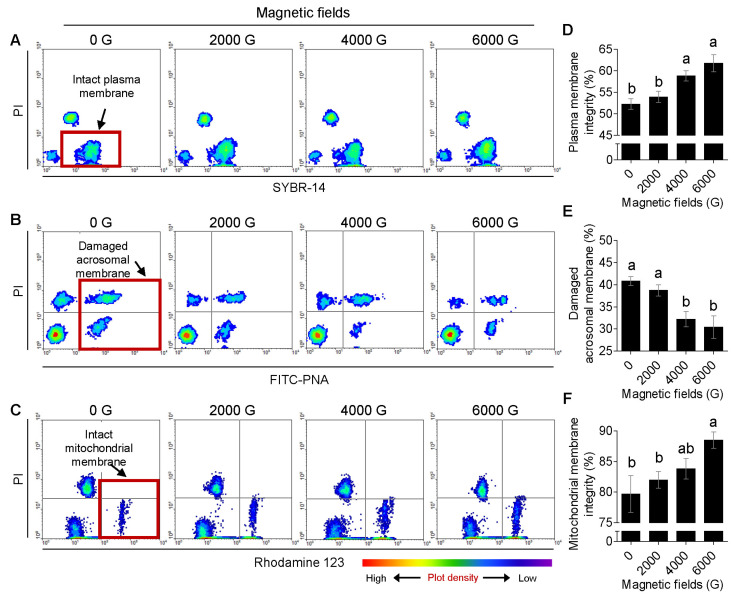
Flow cytometric density-plot of the intact plasma membrane (**A**), damaged acrosomal membrane (**B**), intact mitochondrial membrane (**C**) populations, changes of magnetized freezing extender on plasma membrane integrity (**D**), damaged acrosomal membrane (**E**), and mitochondrial membrane integrity (**F**) in frozen-thawed boar sperm (**B**), ^a,b^
*p* < 0.05.

**Figure 3 animals-13-00634-f003:**
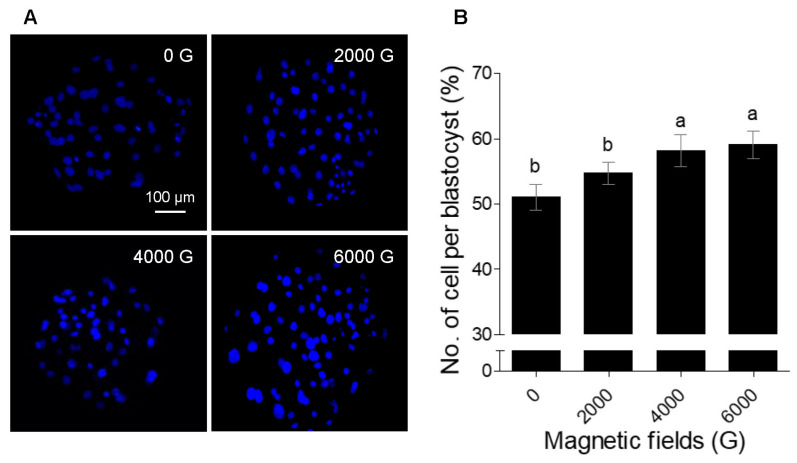
Blastocyst cell number at 168 h after IVF with frozen-thawed boar spermatozoa in the magnetized freezing extender treated with 0 G (*n* = 14), 2000 G (*n* = 16), 4000 G (*n* =15) and 6000 G (*n* = 17), representative images of stained by Hoechst 33342 (**A**) and number of cell per blastocyst (**B**), ^a,b^
*p* < 0.05.

**Table 1 animals-13-00634-t001:** Changes of frozen-thawed boar sperm motility using magnetized freezing extender.

Magnetic Field (G)	TM (%)	PM (%)	VAP (μm/s)	VSL (μm/s)	VCL (μm/s)
0	33.1 ± 5.2 ^b^	17.0 ± 2.7	131.8 ± 8.8	60.3 ± 6.1	70.1 ± 6.8
2000	35.4 ± 5.1 ^b^	20.1 ± 3.6	138.3 ± 8.6	60.4 ± 2.2	71.6 ± 9.8
4000	36.2 ± 2.7 ^b^	20.8 ± 3.0	130.5 ± 11.7	55.2 ± 5.9	74.2 ± 8.3
6000	42.1 ± 2.1 ^a^	21.6 ± 1.9	136.5 ± 10.5	52.1 ± 11.9	78.7 ± 2.9

TM: Total motility, PM: progressive motility, VAP: velocity average path, VSL: velocity straight line, and VCL: curvilinear velocity. Data are presented as mean ± SEM. ^a,b^
*p* < 0.05, *n* = 5.

**Table 2 animals-13-00634-t002:** Development of porcine oocytes after in vitro fertilization with frozen boar sperm by a magnetized freezing extender.

Magnetic Field (G)	No. of Total Oocytes	No. of Developed Embryos from Oocytes (%)	No. of Developed Blastocysts from Oocytes (%)
0	216	146 (68.1 ± 1.2)	25 (11.5 ± 1.1) ^b^
2000	217	152 (70.4 ± 1.4)	40 (18.7 ± 2.2) ^ab^
4000	217	160 (73.4 ± 4.2)	48 (22.0 ± 0.5) ^a^
6000	220	153 (70.2 ± 2.0)	40 (17.5 ± 1.9) ^ab^

Embryos and blastocysts were observed 168 h after in vitro fertilization. Data are presented as mean ± SEM. ^a,b^
*p* < 0.05, *n* = 4.

## Data Availability

The original contributions presented in the study are included in the article, further inquiries can be directed to the corresponding author.

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
