# Peer review of "Effect of Magnetized Freezing Extender on Membrane Damages, Motility, and Fertility of Boar Sperm Following Cryopreservation"

_animals, 2023, doi:10.3390/ani13040634_

Round 1

Reviewer 1 Report

The authors investigated the effects of magnetizing the freezing extenders prior to cryopreservation of boar semen on the post-thaw sperm integrity and fertilizing ability. The information from the research may be of value; however, it was very difficult to understand the manuscript. English language was very poor and entensive editing of the is required.

Much better description of material of methods is required. 

Author Response

Q. The authors investigated the effects of magnetizing the freezing extenders prior to cryopreservation of boar semen on the post-thaw sperm integrity and fertilizing ability. The information from the research may be of value; however, it was very difficult to understand the manuscript. English language was very poor and entensive editing of the is required.

A. Thank you for your specific comments. We checked English grammar of this manuscript using EDITAGE English editing service. We attached certificate of Editing certificate. Additionally, we double checked English language before resubmission. You can find edited sentences (highlight as red) in revised manuscript.

Q. Much better description of material of methods is required.

A. We revised and added materials and methods. Please check red color in revised manuscript.

Reviewer 2 Report

This study aimed to investigate the effect of the magnetized freezing extender on sperm characteristics and the development of oocytes in vitro fertilized with cryopreserved boar semen.

The manuscript is well written and very clear presented, but I have a few suggestions that could help improve it.

Line 77_When you said “terminal crosses of Duroc “ it is not clear what the genotype of the boar is (50%Duroc and 50% other breed, or?). Please, be more clear.

I guess that the boars are not related. Please, indicate.

Line81_ What motility do you mean, how is it estimated? Please, be more clear.

Line195_ Did you test for homogeneity of variance before applying analysis of variance? Please, indicate.

Line249_ Why is n=4 here? Please explain.

Author Response

General comment.

This study aimed to investigate the effect of the magnetized freezing extender on sperm characteristics and the development of oocytes in vitro fertilized with cryopreserved boar semen. The manuscript is well written and very clear presented, but I have a few suggestions that could help improve it.

A. Thank you for your comment. We edited points following three comments as red color. Please check red color in revised manuscript.

Q. Line 77_When you said “terminal crosses of Duroc “ it is not clear what the genotype of the boar is (50%Duroc and 50% other breed, or?). Please, be more clear.

A. We used purebred Duroc (Duroc x Duroc). We changed terminal crosses of Duroc to Duroc in revised manuscript (L85-86).

Q. I guess that the boars are not related. Please, indicate.

A. We changed boars to pigs in revised manuscript (Line 84). Thank you for your specific comments.

Q. Line81_ What motility do you mean, how is it estimated? Please, be more clear.

A. We used sperm samples which are more than 80% total motility. We revised this contents in revised manuscript (Line 90).

Q. Line195_ Did you test for homogeneity of variance before applying analysis of variance? Please, indicate.

A. We test homogeneity of variance before statistical analysis. We added this contents in revised manuscript (Line 203-205). Thank you for your specific comments.

Q. Line249_ Why is n=4 here? Please explain.

A. We used 50-60 embryos for one replication experiment. In results, total 216 to 220 per treatment were used in investigation of in vitro fertility of frozen-thawed sperm. We added this contents in revised manuscript (Line 193).

Reviewer 3 Report

Introduction:

The introduction is complete and adequate, but the formation of ROS is justified as one of the main causes of damage to freezing, an effect that the authors have not evaluated experimentally. It's part should be reduced.

Material and methods:

In sperm freezing, they should explain the freezing process better. It is not clear how they did the freezing, whether using nitrogen vapors, or a biofreezer. Specify the method further. Since the freezing curve is essential for determining how the ice crystals have formed.

Why has that thawing rate been chosen? Speed can cause recrystallization at some points in the thaw. Indicate justifiably why this ramp has been used.

Discussion:

line 270. The authors cannot assure this "This means that the increased electronic donor of the magnetized freezing extender protected against cryo-damage by ROS during freezing" since the authors have not done tests to determine if the ROS substances increase or decrease in the cellular interior, or tests that determine lipid peroxidation caused by ROS or an increase or decrease in antioxidant capacity.

The authors state that “The final size, shape, and distribution of the resultant ice crystals are dependent on 284 the cooling rate, degree of supercooling, and nucleation temperature [15,34,35].” The authors must specify why they have chosen the freezing ramp they have used, if it is based on bibliography or if they have done tests before, since it is essential for the justification of the work.

References:

Reference 20, is the title in capital letters.

Author Response

Introduction:

Q. The introduction is complete and adequate, but the formation of ROS is justified as one of the main causes of damage to freezing, an effect that the authors have not evaluated experimentally. It's part should be reduced.

A. We deleted contents regarding of ROS in introduction section. Please check revised manuscript.

Material and methods:

Q. In sperm freezing, they should explain the freezing process better. It is not clear how they did the freezing, whether using nitrogen vapors, or a biofreezer. Specify the method further. Since the freezing curve is essential for determining how the ice crystals have formed.

A. We did the freezing 18°C to 5°C during 120 min using simple sperm cooling method. Detail method is that diluted experimental tube containing sperm with freezing extender put into 500 mL of 18°C water and then, they was cooled in -18°C freezer during 120 min. This contents were added in revised manuscript (Line 123-125).

Q. Why has that thawing rate been chosen? Speed can cause recrystallization at some points in the thaw. Indicate justifiably why this ramp has been used.

A. We know that there are many thawing conditions for frozen boar sperm. Before this study, we tested various thawing condition for frozen boar sperm (data not shown) in previous our study (Reprod Domest Anim 2019, 54, 1251-1257), in results, we confirmed that 45s and 38°C are best condition for thawing of frozen boar sperm and choose 45s and 38°C in this study.

Discussion:

Q. line 270. The authors cannot assure this "This means that the increased electronic donor of the magnetized freezing extender protected against cryo-damage by ROS during freezing" since the authors have not done tests to determine if the ROS substances increase or decrease in the cellular interior, or tests that determine lipid peroxidation caused by ROS or an increase or decrease in antioxidant capacity.

A. We removed contents regarding of ROS in discussion section of revised manuscript. We are scheduled to investigate about influence of magnetized freezing extender on ROS in frozen-thawed boar sperm in the future. Thank you for specific comment.

Q. The authors state that “The final size, shape, and distribution of the resultant ice crystals are dependent on 284 the cooling rate, degree of supercooling, and nucleation temperature [15,34,35].” The authors must specify why they have chosen the freezing ramp they have used, if it is based on bibliography or if they have done tests before, since it is essential for the justification of the work.

A. We added the contents about freezing ramp following your comment in revised manuscript (Line 293-296) and added new references of relation between freezing rate and sperm viability, motility in frozen-thawed boar sperm. Thank you for your specific comments.

References:

Q. Reference 20, is the title in capital letters.

A. We revised reference 20.
